# Synthesis and Structure of Iron (II) Complexes of Functionalized 1,5-Diaza-3,7-Diphosphacyclooctanes

**DOI:** 10.3390/molecules25173775

**Published:** 2020-08-19

**Authors:** Yulia S. Spiridonova, Yulia A. Nikolaeva, Anna S. Balueva, Elvira I. Musina, Igor A. Litvinov, Igor D. Strelnik, Vera V. Khrizanforova, Yulia G. Budnikova, Andrey A. Karasik

**Affiliations:** Arbuzov Institute of Organic and Physical Chemistry, FRC Kazan Scientific Center, Russian Academy of Sciences, Arbuzov str. 8, 420088 Kazan, Russia; aik79@iopc.ru (Y.S.S.); nikolaeva@iopc.ru (Y.A.N.); elli@iopc.ru (E.I.M.); litvinov@iopc.ru (I.A.L.); igorstrelnik@iopc.ru (I.D.S.); KhrizanforovaVera@yandex.ru (V.V.K.); yulia@iopc.ru (Y.G.B.); karasik@iopc.ru (A.A.K.)

**Keywords:** 1,5-diaza-3,7-diphosphacyclooctane, iron, complex, synthesis, structure, redox properties

## Abstract

In order to synthesize new iron (II) complexes of 1,5-diaza-3,7-diphosphacyclooctanes with a wider variety of the substituents on ligands heteroatoms (including functionalized ones, namely, pyridyl groups) and co-ligands, it was found that these ligands with relatively small phenyl, benzyl, and pyridin-2-yl substituents on phosphorus atoms in acetonitrile formed bis-P,P-chelate *cis*-complexes [L_2_Fe(CH_3_CN)_2_]^2+^ (BF_4_)_2_^−^, whereas P-mesityl-substituted ligand formed only monoligand P,P-complex [LFe(CH_3_CN)_4_]^2+^ (BF_4_)_2_^−^. 3,7-dibenzyl-1,5-di(1′-(*R*)-phenylethyl)-1,5-diaza-3,7-diphosphacyclooctane reacted with hexahydrate of iron (II) tetrafluoroborate in acetone to give an unusual bis-ligand cationic complex of the composition [L_2_Fe(BF_4_)]^+^ BF_4_^−^, where two fluorine atoms of the tetrafluoroborate unit occupied two pseudo-equatorial positions at roughly octahedral iron ion, according to X-ray diffraction data. 1,5-diaza-3,7-diphosphacyclooctanes replaced tetrahydrofurane and one of the carbonyl ligands of cyclopentadienyldicarbonyl(tetrahydrofuran)iron (II) tetrafluoroborate to form heteroligand complexes [CpFeL(CO)]^+^BF_4_^−^. The structural studies in the solid phase and in solutions showed that diazadiphosphacyclooctane ligands of all complexes adopted chair-boat conformations so that their nitrogen atoms were in close proximity to the central iron ion. The redox properties of the obtained complexes were performed by the cyclic voltammetry method.

## 1. Introduction

The complexes of aminomethylphosphines with earth-abundant metals, such as nickel, cobalt, and iron, draw permanent attention as molecular catalysts and electrocatalysts for the activation of small molecules (H_2_, N_2_, CO_2_, etc.) [1,2,3,4,5,6]. The metal complexes of cyclic and acyclic aminomethylphosphines-containing amine centers in the main framework of a diphosphine ligand as proton relay act as the mimetics of the [FeFe]- and [NiFe]-hydrogenases, which are fast and efficient catalysts for the oxidation/generation of H_2_ [1,2,3,4]. The most studied objects are the nickel (II) complexes of 1,5-diaza-3,7-diphosphacyclooctanes, which show the high effectiveness in the electrocatalysis, both of the reduction of protons to generate H_2_ and of the oxidation of H_2_ [1,2,3,7], and have been successfully used as both the cathode and anode catalysts in fuel cells [4,8,9]. A very wide variation of the substituents on the ligands heteroatoms has been performed for these nickel-containing complexes in order to find the most effective and convenient catalysts for different applications [2,7,8,10,11,12,13].

It has been shown that the nature of the central ion could essentially influence on the type of the complexes formed on the basis of 1,5-diaza-3,7-diphosphacyclooctanes. In particular, the interaction of these ligands with chromium dichloride leads to the unexpected ring expansion and the formation of the chromium (II) complexes of 12-membered P_3_N_3_ and 16-membered P_4_N_4_ macrocyclic aminomethylphosphines [14]. Another example of the ring expansion of 1,5-di-*p*-tolyl-3,7-di(pyridine-2′-yl)-1,5-diaza-3,7-diphosphacyclooctane on Au_3_ cluster into 16-membered P_4_N_4_ macrocycle has been observed [15]; it should be mentioned that this ligand transformation does not take place, both for other diazadiphosphacyclooctane ligands [15] and for other gold cores [16,17]. These data indicate that the formation of the metal complexes of diazadiphosphacyclooctanes cannot be considered a completely predictable process.

The iron complexes of 1,5-diaza-3,7-diphosphacyclooctanes are less studied in spite of their promising activity in H_2_ splitting and electrooxidation [3,18,19,20] and N_2_ binding and activation [5]. In particular, the set of substituents on phosphorus atoms of the investigated P_2_N_2_ ligands is restricted and includes phenyl [21,22,23,24], *tert*-butyl [19,20,24,25], cyclohexyl [21,24], and benzyl groups [24], whereas the nitrogen atoms bear phenyl or benzyl groups. These diphosphines are used for the synthesis of bis-ligand iron (II) complexes [21,23] or more often for the preparation of heteroligand complexes containing cyclopentadienyl [19,20,22,25] or benzenedithiolate ligands [24]. The cyclopentadienyl-containing complexes draw the main interest due to their activity in hydrogen electrooxidation [3,19,20,25]. The only additionally functionalized diazadiphosphacyclooctane ligand, namely, 1,5-diphenyl-3,7-bis(2′-diphenylphosphinoethyl)-1,5-diaza-3,7-diphosphacyclooctane, has been used for the synthesis of unusual square planar iron complexes stabilized by the coordination with exocyclic phosphino groups, which are able to activate both hydrogen and nitrogen molecules [5,18].

It should be mentioned that the varying of the substituents on the nitrogen atoms of acyclic P_2_N ligands, namely, bis(diethylphosphinomethyl)organylamines, has shown the essential influence of the nature of these substituents on the activity of the corresponding iron (II) complexes in the hydrogen electrooxidation [26]. These data indicate that the wider variety of the diazadiphosphacyclooctane ligands of iron complexes and the further study of their structures and properties is expedient for the search of new catalysts on the basis of these compounds.

The synthesis of various types of homo- and heteroligand new iron (II) complexes of 1,5-diaza-3,7-diphosphacyclooctanes, including additionally functionalized P-pyridyl substituted ligands, and the results of our study of their structures and redox properties were reported in this contribution.

## 2. Results

1,5-Diaza-3,7-diphosphacyclooctanes **1**–**4** with relatively small substituents on phosphorus atoms, namely, phenyl, benzyl, and pyridin-2-yl groups, reacted with hexa(acetonitrile)iron (II) tetrafluoroborate in acetonitrile to give yellow or orange bis-ligand complexes [L_2_Fe(CH_3_CN)_2_]^2+^ (BF_4_)_2_^−^
**6**–**9**, according to the data of elemental analysis. In ESI or MALDI mass-spectra of all these complexes, the peaks of ions containing two diazadiphosphacyclooctane moieties, iron ion, and in some cases, various co-ligands were observed. ^31^P-NMR spectra of complexes **6**–**9** showed two complex second-order multiplets of the equal intensity in the ranges 25.6–34.7 and 34.2–41.2 ppm. This spectral pattern is typical for AA’BB’ spin system of *cis*-iron (II) complexes with the strongly distorted octahedral configuration of the central ion-containing two P,P-chelating diphosphine ligands and two acetonitrile co-ligands, which occupy formally equatorial positions [21], so it indicated the analogous *cis*-structures of the complexes **6**–**9** (Scheme 1). Their ^1^H-NMR spectra also indicated the highly asymmetric structures of the complexes. The substituents on phosphorus atoms were mutually nonequivalent; the substituents on nitrogen atoms provided two sets of the corresponding signals, which probably corresponded to the chair-boat conformation of the cyclic diphosphine ligands so that two halves of the ligand molecule were in the different spatial environment (Figure 1). One of the nitrogen atom (and the corresponding substituent) of each ligand was in close proximity to the second diazadiphosphacyclooctane ligand, whereas the other nitrogen atom and its substituent were located on the side of acetonitrile ligand and took the remote positions. The very complex spectral patterns observed for the protons of P-CH_2_-N fragments were in accordance with these conformations. 

The singlets of methyl groups of coordinated acetonitrile molecules were observed near 1.96 ppm in the spectra of complexes **6**–**9**. It indicated that the exchange between coordinated and uncoordinated acetonitrile molecules was not fast. However, it should be mentioned that all complexes **6**–**9** were sensitive to the solvent nature, and their partial decomposition was observed in dimethylformamide (DMF), alcohols, and even in acetone or dichloromethane. In the last solvents, the decomposition was probably caused by the presence of traces of water or hydrogen chloride, which led to the partial substitution of acetonitrile co-ligands, so these ligands might be considered as relatively labile. The solvent sensitivity of these complexes restricted the possibility of their variable-temperature NMR studies.

The proposed structure of the complex **8** was confirmed by X-ray diffraction analysis. A single crystal of **8** was grown by the gas-phase diffusion of acetonitrile to the solution of **8** in methylene chloride. The iron ion had a distorted octahedral configuration (Figure 2a). The acetonitrile ligands occupied the formally equatorial positions. As a result, bonds P7-Fe and P7B-Fe were formally apical (a bond angle P7-Fe1-P7B is 175.4(1)°), and the other P-Fe bonds were formally equatorial. The selected bond lengths and angles of complex **8** are presented in Table 1. There was not a noticeable difference between the lengths of apical bonds Fe1-P7, Fe1-P7B (2.232(3) and 2.242(3) Å) and equatorial bonds Fe1-P3, Fe1-P3B (2.233(3) and 2.246(3) Å). The equatorial bond lengths N40-Fe1 and N43-Fe1 were 1.960(10) and 1.940(9) Å, the bond angle N40-Fe1-N43 was only 82.9(4)°, whereas the bond angle P3-Fe1-P3B was enlarged up to P3-Fe1-P3B 104.09(1)°. The similar structure was observed for the previously described *cis*-[bis(1,5-dibenzyl-3,7-diphenyl-1,5-diaza-3,7-diphosphacyclooctane)di(acetonitrile)-iron(II)] tetrafluoroborate [21].

Both diazadiphosphacyclooctane ligands (N(1)P(3)N(5)P(7) and N(1B)P(3B)N(5B)P(7B)) had chair-boat conformations, so the substituents on nitrogen atoms in the chair and boat parts of the heterocycles were in an essentially different environment. That explained the nonequivalence of these substituents in NMR spectra. It should be mentioned that the high asymmetry of the complex and the nonequivalence of two halves of the ligand cycle had to take place even in the case of conformational exchange between two chair-boat conformations of diazadiphosphacyclooctane. The endocyclic nitrogen atoms of the boat parts of both ligands were in the proximity to the iron ion (the distances Fe…N5 and Fe…N5B are 3.552(8) and 3.566(8) Å respectively). This arrangement was favorable for the secondary interactions, similar to ones observed in natural hydrogenases [3].

The interaction of hexa(acetonitrile)iron (II) tetrafluoroborate with 1,5-diaza-3,7-diphosphacyclooctane **5** bearing bulky mesityl groups on phosphorus atoms led only to monoligand P,P-chelate complex [LFe(CH_3_CN)_4_]^2+^ (BF_4_)_2_^−^
**10** (L = **5**) (Scheme 1), even in the presence of the excess of the ligand. The exclusive formation of complexes with metal-ligand ratio 1:1 has been earlier shown for 1,5-diaza-3,7-diphosphacyclooctane with bulky substituents on phosphorus atoms and other transition metal (Pt (II), Pd (II), Ni(II)) derivatives [7,27,28,29]. Both MALDI and ESI mass-spectra of **10** showed a peak with *m*/*z* 621 corresponding to [LFe]^+^ ion (L = **10**). Only one narrow signal at δ_P_ 55.56 ppm was observed in its ^31^P-NMR spectrum. Unlike the spectra of the complexes **6–9**, the ^1^H-NMR spectrum of **10** showed only one set of signals for every group of ligand’s protons. The spectral pattern was typical for the chair-boat conformation of diazadiphosphacyclooctane ligand in the case of the fast conformational exchange in the NMR time scale in the solutions [29]. The absence of the signal of coordinated acetonitrile molecules indicated also the fast exchange and the lability of acetonitrile co-ligands.

The acetonitrile co-ligands are considered as ones, which form relatively strong coordination bonds with iron (II) ion [21], so in order to obtain the iron complex with less coordinating co-ligands or even coordinationally unsaturated iron complex of 1,5-diaza-3,7-diphosphacyclooctane, the hydrate iron complex was used as the starting reagent in a low-coordinating solvent. Ligand **3** readily reacted with iron (II) tetrafluoroborate hexahydrate [Fe(H_2_O)_6_](BF_4_)_2_ in acetone to give a mixture of two bis-ligand *cis*-complexes **11a** and **11b** (Scheme 1). In the ^31^P-NMR spectrum of the reaction mixture, two pairs of doublets at δ_P_ 38.21 and 31.03 ppm (^2^*J*_PP_ ≈ ^2^*J*_PP_ ≈ 70.7 Hz) (**11a**), 37.30 and 29.94 ppm (^2^*J*_PP_ ≈ ^2^*J*_PP_ ≈ 71.8 Hz) (**11b**) with a ratio of intensities of 1.4:1 were observed. Nevertheless, after the concentration of the reaction mixture and the crystallization of the residue at −15 °C, only the minor product **11b** was isolated as small dark-red crystals in the yield of 41%. The elemental analysis data of **11b** corresponded to the composition L_2_Fe(BF_4_)_2_ (L = **3**). Its mass-spectrum showed peaks with *m*/*z* 1132 and 1162 corresponding to the ions [L_2_Fe]^+^ and [L_2_FeBF]^+^. Unexpectedly, the bands of hydroxyl groups were absent in the IR-spectrum of **11b**, whereas the characteristic very strong broadband of BF_4_ anion was observed at 1057 cm^−1^. Two doublets at 37.30 and 29.94 ppm (^2^*J*_PP_ = ^2^*J*_PP_ = 71.8 Hz) were observed in its ^31^P-NMR spectrum. Like the spectra of complexes **6**–**9**, the ^1^H-NMR spectrum of **11b** demonstrated the pairwise nonequivalence of all protons of the heterocyclic fragments, of the benzylic methylene groups, and 1-phenylethyl substituents on nitrogen atoms, which indicated the chair-boat conformations of diazadiphosphacyclooctane rings. The absence of the signals of the coordinated water in the spectrum was in accordance with the data of IR-spectroscopy and confirmed the absence of water co-ligands in the coordination sphere of the iron. 

The structure of the complex **11b** was finally established by X-ray diffraction analysis. Unfortunately, despite numerous attempts to grow a crystal of satisfactory quality, it was not possible, and the crystal structure of the compound was established with large experimental errors; nevertheless, the structure of the complex cation was established reliably. Therefore, in this work, we presented only the conformation and selected geometry parameters of the cation, but not the crystal structure of this compound. The dark-red single crystal of **11b** was grown by the gas-phase diffusion of acetone to the solution of **11b** in methylene chloride. It should be mentioned that the crystal is an acetone solvate and contains an undetermined solvent. Molecule **11b** appeared to be a bis-ligand monocationic complex of the composition [L_2_Fe(BF_4_)]^+^ BF_4_^−^, where two fluorine atoms of one of the tetrafluoroborate unit were included in the coordination sphere of the iron ion and occupied two pseudo-equatorial positions (Figure 2b). A similar coordination mode was described for the complexes of bis- and tetrakis-phosphines with iron (II) sulfate and carbonate [30,31], but for the low-coordinating tetrafluoroborate anion, it was unexpected. The molecule **11b** in this crystal was on the special position on the two-fold axis 2 (atoms Fe and B lied on the two-fold axis 2). In the unit cell of the crystal, 888 Å^3^ free volumes remained, in which other solvate molecules could not be identified. As a result, the refinement of the structure was carried out using the SQUEEZE procedure. It should be mentioned that a simultaneous TG/DSC analysis of the single crystals of **11b** showed a loss of two water molecules and a half of acetone molecule per one molecule of the complex in the temperature range 60–138 °C (Appendix A), so the undetermined disordered solvent was the water and acetone.

The central iron ion had the strongly distorted octahedral configuration: the dihedral angle between P(7)-Fe-P(7′) and F(1)-Fe-F(1′) planes was 20(1)°, and the bond angles P(7)-Fe-P(7′) and F(1)-Fe-F(1′) were 101.1(2) and 70.4(4)°, respectively, (P(7′) and all atoms with ‘ were symmetry-related atoms, symmetry operation 2 − x, 1 − x + y, 1 − z). The bond angle P(3)-Fe-P(3′) was 176.5(2)°. P,P-chelating diazadiphosphacyclooctane ligands had chair-boat conformations; phosphorus atoms of every ligand coordinated the iron atom in pseudo-equatorial and pseudo-apical positions. Bond lengths of equatorial bonds Fe1-P7 were 2.209(4), and the axial bonds Fe1-P3 were 2.267(3) Å (the selected bond lengths and angles of complex **11b** are presented in Table 1). Nitrogen atoms of the boat parts of the heterocycles were directed to the coordinated tetrafluoroborate fragment so that they were in the proximity, both to the central ion (the shortest distance Fe1…N5 was 3.314(11) Å) and to the coordinated fluorine atoms (the shortest distance F…N was only 2.59(2) Å and less than a sum of van der Waals radii). Their lone electron pairs were directed to the metal iron. This structure might be favorable for the small molecules activation due to the secondary interactions of the metal-coordinated substrates with amine centers of the ligands. The comparison of the structural parameters of complexes **8** and **11b** showed more noticeable distortion of the octahedral geometry of iron ion in **11b**, namely, the small value of F1-Fe-F1′ bond angle and some lengthening of apical Fe–P bonds in comparison with equatorial ones (2.267(3) vs. 2.209(4) Å), whereas Fe–P distances of complex **8** varied in the narrow range of 2.232(3)–2.246(3) Å (Table 1). The bond lengths Fe–F were close (2.050(9) Å). The bonds B–F with the coordinated fluorine atoms were slightly longer than the corresponding bonds with uncoordinated ones (1.49(2) Å vs. 1.39(2) Å). The Fe–F bond lengths were within the typical range of few known BF_4_-ligated iron (II) complexes (1.96–2.26 Å), which contained only terminally η^1^-coordinated tetrafluoroborate anions [32,33,34,35,36,37,38,39]. The most of known complexes-containing bidentate κ^2^-bound tetrafluoroborate ligands were silver (I) complexes with weakly coordinated BF_4_ anions, where the Ag–F distances were relatively long (2.79–3.01 Å) [40,41,42,43]. Only two examples of an almost covalent κ^2^-F_2_BF_2_ bonding of tetrafluoroborate anion had been described, namely, the silver (I) complex of cyclodextrine-based diphosphine, where BF_4_^−^ counterion lied deep inside the cavity, and Ag–F distances were only 2.53–2.62 Å [44], and dimolibdenum complex [Mo_2_Cp_2_(k^2^-F_2_BF_2_)(μ-PPh_2_)_2_(CO)](BF_4_), which showed close and short Mo–F distances (2.18(1) Å) [45]. These values were quite close to the metal-fluorine distances measured for related η^1^-FBF_3_ coordinated tungsten and molybdenum complexes [45]. According to these structural criteria, namely, the usual lengths of two Fe–F bonds in the cation of **11b** and a small difference between them, the Fe/BF_4_ interaction might be considered as relatively strong and close to the covalent one. 

The complex **11a** was not isolated, but most probably it was the corresponding dihydrate complex with uncoordinated tetrafluoroborate anions; the easy replacement of two water ligands by bidentate sulfate or carbonate ligand from the outer sphere was described for the iron (II) complexes of 1,5-di[di(hydroxymethyl)phosphinomethyl]-3,7-di(hydroxymethyl)-1,5-diaza-3,7-diphosphacyclooctane [30].

The attempt to register NMR spectra of **11b** in CD_3_CN showed the fast change of the solution color from red to yellow and the appearance of the signals of corresponding bis-acetonitrile complex **8** (multiplets at 25.6 and 34.2 ppm) along with other signals in close ranges, which presumably corresponded to the products of the incomplete ligand replacement. The possibility of the exchange of tetrafluoroborate ligand to acetonitrile led to the attempt to synthesize complex **8** directly from the more accessible and less sensitive iron (II) tetrafluoroborate hexahydrate. Its interaction with the ligand **3** in acetonitrile indeed led finally to the complex **8**, but the reaction was very slow, and its completion required about 50 days vs. 12 h in the case of starting from [Fe(CH_3_CN)_6_](BF_4_)_2_, so this route was inadvisable.

The interaction of P-benzyl-substituted ligand **3** with cyclopentadienyldicarbonyl(tetrahydrofuran)iron (II) tetrafluoroborate in toluene at ambient temperature for 1 day smoothly led to the replacement of tetrahydrofuran (THF) and one of the carbonyl ligand by the diphosphine ligand **3** and to the formation of cationic complex **12** (Scheme 1) in a yield of 82%. A similar complex containing chloride counter-ion was isolated in relatively low yield (20%) during the early stages of the photolysis of a mixture of CpFe(CO)_2_Cl and 1,5-dibenzyl-3,7-diphenyl-1,5-diaza-3,7-diphosphacyclooctane, whereas the final and main product of the photolysis was the complex CpFe(P^Ph^_2_N^Bn^_2_)Cl [22]. The reaction of the ligand **4** with [CpFe(CO)_2_(THF)]BF_4_^−^ proceeded slowly, probably due to a decreased nucleophilicity of phosphorus atoms bearing relatively electron-withdrawing pyridyl substituents. The completion of the reaction required the heating of the reaction mixture at 60–70 °C for 4 days, and the formation of the analogous complex **13** (Scheme 1) was complicated by the precipitation of a large amount of low-soluble and paramagnetic dark resin, so the yield of **13** was only 9%. Presumably, the prolonged heating led to a loss of its carbonyl ligand. According to elemental analysis data, the complexes **12** and **13** had the composition [CpFeL(CO)]^+^BF_4_^−^, and their ESI mass-spectra showed the peaks of corresponding cations [CpFeL(CO)]^+^ (L = **3**, **4**) with *m*/*z* 687.3 and 633.4, respectively. Two bands of the carbonyl group vibrations at 1940 and 1950–1951 cm^−1^ were observed in the IR spectra of **12** and **13**. In the ^31^P-NMR spectrum of **13**, one narrow signal at 59.14 ppm was observed, whereas the spectrum of **12** showed two close doublets of AB system at 55.55 and 56.27 ppm (^2^*J*_PP_ = 116.4 Hz) because two phosphorus atoms were nonequivalent due to the presence of the chiral substituents on nitrogen atoms [46]. In the ^1^H-NMR spectrum of **13**, the nonequivalence of the *p*-tolyl substituents on nitrogen atoms and the pairwise nonequivalence of the methylene groups in the heterocyclic fragment were observed as a result of the chair-boat conformation of the diazadiphosphacyclooctane ligand. In the ^1^H-NMR spectrum of **12**, the presence of chiral substituents on nitrogen atoms led additionally to the nonequivalence of benzyl substituents on phosphorus atoms and to the nonequivalence of all four methylene groups of the chair-boat diazadiphosphacyclooctane fragment, which formed complex overlapped multiplets. The protons of cyclopentadienyl ligands were registered as a singlet at 5.18 ppm for **12** and as a triplet at 5.05 ppm (^2^*J*_PH_ = 1.4 Hz) for **13**. 

The redox properties of synthesized iron complexes **6**–**13** were studied by the cyclic voltammetry in MeCN. The results are presented in Table 2 and Figure 3. One reduction and oxidation peaks were observed for all iron complexes. One irreversible reduction peak, as we supposed, corresponding to FeII/I couple, was observed on cyclic voltammograms (CVs) at range from −1.37 to −1.74 V (for complexes **6**–**10**) and depended on substituents at phosphorus and nitrogen atoms of ligands. The reversible oxidation peak corresponding FeII/III couple was observed at 0.25–0.34 V in these cases. In the case of complexes **12** and **13**, slightly negative shift to −1.60–1.80 V of reduction potential (vs. bis-chelated complexes **6**–**9**) and irreversible oxidation peaks at 1.10–1.23 V were observed on CVs. 

## 3. Discussion

1,5-Diaza-3,7-diphosphacyclooctanes **1**–**4** with relatively small phenyl, benzyl, and pyridin-2-yl substituents on phosphorus atoms reacted with hexa(acetonitrile)iron (II) tetrafluoroborate in acetonitrile as P,P-chelating ligands to give bis-ligand *cis*-complexes [L_2_Fe(CH_3_CN)_2_]^2+^ (BF_4_)_2_^−^ (**6**–**9**), whereas the P-mesityl-substituted ligand **5** formed only monoligand P,P-complex [LFe(CH_3_CN)_4_]^2+^ (BF_4_)_2_^−^ (**10**). The interaction of the ligand **3** with hexahydrate of iron (II) tetrafluoroborate in acetone unexpectedly led to a bis-ligand cationic complex of the composition [L_2_Fe(BF_4_)]^+^ BF_4_^−^ (**11b**), where two fluorine atoms of one of the tetrafluoroborate unit were included into the coordination sphere of the iron ion and occupied two pseudo-equatorial positions. 1,5-diaza-3,7-diphosphacyclooctane ligands were able to replace tetrahydrofurane and one of the carbonyl ligands of cyclopentadienyldicarbonyl(tetrahydrofuran)iron (II) tetrafluoroborate to form heteroligand complexes **12** and **13** of the composition [CpFeL(CO)]^+^BF_4_^−^. The diazadiphosphacyclooctane ligands of all complexes adopted chair-boat conformations so that their nitrogen atoms were in close proximity to the central iron ion. In spite of the favorable disposition of the metal center and pendant amine centers, the redox properties of the obtained complexes (in particular, the presence of the irreversible reduction peaks) were not promising for the application of these complexes as electrocatalysts. 

## 4. Materials and Methods

### 4.1. General

All the reactions and manipulations with phosphines **1**–**5** were carried out under a dry argon atmosphere using standard vacuum-line techniques. All manipulations with complexes **6**–**13**, excluding the electrochemical measurements, were carried out under normal conditions without inert atmosphere. Solvents were purchased from Acros Organics (Geel, Belgium) and were purified, dried, deoxygenated, and distilled before use. Iron (II) tetrafluoroborate hexahydrate, hexa(acetonitrile)iron (II) tetrafluoroborate and cyclopentadienyldicarbonyl(tetrahydrofuran)iron (II) tetrafluoroborate were purchased from Sigma-Aldrich (St. Louis, MO, USA) and used as received.

ESI_pos_ MS were recorded with an AmazonX (Bruker Daltonics GmbH, Bremen, Germany) spectrometer at a capillary voltage of 4500 V. DataAnalysis 4.0 (Bruker Daltonics GmbH, Bremen, Germany) program was used to process the mass spectrometry data. MALDI MS was recorded with an Ultraflex III TOF/TOF (Bruker Daltonics, Germany) spectrometer on the *p*-nitroaniline (pNA) matrix. FlexAnalysis 3.0 (Bruker Daltonics GmbH, Bremen, Germany) program was used to process the mass spectrometry data. The mass spectra are reported as *m*/*z* values. ^1^H-NMR (400 MHz and 600 MHz) and ^31^P-NMR (162 and 242 MHz) spectra were obtained with Bruker Avance-DRX 400 (BrukerBioSpin, Billerica, MA, USA) and Bruker Avance-600 spectrometers (BrukerBioSpin, Billerica, MA, USA). The chemical shifts are reported in ppm relative to SiMe_4_ (^1^H, internal standard) and 85% H_3_PO_4_ (aq) (^31^P, external standard). The coupling constants (*J*) are reported in Hz. The IR spectra were obtained on a spectrometer “Vector-22” (Bruker Optics GmbH, Bremen, Germany) in the range 400–4000 cm^−1^ in Nujol mulls. Determination of the CHN content was carried out on CHN analyzer “CHN-3 KBA”. The determination of the phosphorus content was provided by combustion in an oxygen stream. Simultaneous thermogravimetry and differential scanning calorimetry (TG/DSC) analysis of samples of **11b** (solvate) (10.7 mg) were performed using the STA 449F1 Jupiter (Netzsch, Selb, Germany) thermoanalyzer (in the range of temperatures from 40 to 200 °C in aluminum crucible under a dynamic atmosphere of argon 75 mL/min). The heating rate was 10 °C/min.

Starting phosphines **1** [47], **2** [48], and **4 [2]** were prepared according to the literature procedures. 

### 4.2. X-ray Crystallography Data

The data of **8** were collected on a Bruker SmartApex II (Bruker AXS GmbH, Karlsruhe, Germany) using the ω-scan mode, and the data of **11b** were collected on a Bruker KappaApex II diffractometer (Bruker AXS GmbH, Karlsruhe, Germany) (λ(MoK_α_) = 0.71073 Å), at temperature 100(2) K. The performance mode of the sealed X-ray tube was 50 kV, 30 mA. A suitable crystal of appropriate dimensions was mounted on a glass fiber in a random orientation. Data collection: images were indexed and integrated using the APEX3 data reduction package (v2018.7-2, Bruker AXS, Madison, WI, USA). Final cell constants were determined by the global refinement of reflections from the complete data set. Data were corrected for systematic errors and absorption using SADABS-2016/2 (v2016-2, Bruker AXS, Madison, WI, USA). XPREP-2014/2 (v2014-2, Bruker AXS, Madison, WI, USA) and the Assign Space group routine of WinGX-2018.3 (v2018-3, Bruker AXS, Madison, WI, USA) were used for the analysis of systematic absences and space-group determination. The structures were solved by the direct method using SHELXT-2018/2 (v2018-2, Bruker AXS, Madison, WI, USA) [49] and refined by the full-matrix least-squares on *F*^2^ using SHELXL-2018/3 (v2018-3, Bruker AXS, Madison, WI, USA) [50] and refined by the full-matrix least-squares on *F*^2^ using SHELXL-2018/3 (v2018-3, Bruker AXS, Madison, WI, USA) [50]. Calculations were mainly performed using the WinGX-2018.3 suite of programs [51]. Non-hydrogen atoms were refined anisotropically. The hydrogen atoms were inserted at the calculated positions and treated as riding atoms. The absolute structure of crystals was confirmed by the Flack parameter [52]. 

#### 4.2.1. Crystal Data and Structure Refinement for Compound **8**

C_72_H_86_FeN_6_P_4_^2+^, 2(BF_4_^−^), 2(C_2_H_3_N)^−^; *M* = 1470.93, orthorhombic, space group *P*2_1_2_1_2_1_, *a* = 13.7178(14), *b* = 20.452(2), *c* = 27.659(3) Å, *V* = 7759.9(14) Å^3^, *Z* = 4, *D*_calc_ = 1.259 g cm^−3^; *μ*(Mo_Kα_) = 0.342 mm^−1^; Θ_Max_ = 25.8°, 134,467 reflections were measured, 14,690 were independent reflections. Final *R*_1_ = 0.0791, *R*_w_ = 0.1537, goodness of fit 1.007 for 7289 reflections with *I* ≥ 2*σ*(*I*), and *R*_1_ = 0.2051, *R*_w_ = 0.2081 for all reflections; flack parameter (absolute structure parameter) 0.006(12).

#### 4.2.2. Crystal Data and Structure Refinement for Compound **11b**

3(C_68_H_68_BF_4_FeN_4_P_4_^+^), 2(BF_4_^−^), 15(C_3_H_6_O), [+ solvent], one BF_4_^−^ anion was not solved; *M* = 4704.49, trigonal, space group *P*321, *a* = *b* = 22.651(2), *c* = 15.3431(17) Å, *V* = 6817.4(16) Å^3^, *Z* = 1, *D*_calc_ = 1.146 g cm^−3^; *μ*(Mo_Kα_) = 0.297 mm^−1^; 29,199 reflections were measured, 8934 were independent reflections. Final *R*_1_ = 0.1040, *R*_w_ = 0.2221 for 4109 reflections with *I* ≥ 2*σ*(*I*), and *R*_1_ = 0.2191, *R*_w_ = 0.2794 for all reflections, goodness of fit 0.978, flack parameter (absolute structure parameter) −0.04(4). Organometallic cation was on a special position on the two-fold axis; BF_4_ anion was on a special position on the trifold axis; molecules disordered on 6 positions BF_4_^−^ anion were disordered by two-fold and trifold axes and were not solved from different Fourier maps, and one of the solvate molecules of acetone was on a special position on the two-fold axis. In this crystal, the total potential solvent-accessible void volume (SOLV-Map Value) 926 Å^3^ was found, but solvate molecules and those disordered on 6 positions of BF_4_^−^ anion were not found from the different Fourier maps of electron density, and the final stages of structure refinement were performed using the SQUEEZE procedure of program PLATON [53].

CCDC 2000581 (**8**), CCDC 2000582 (**11b**) contained the crystallographic data. These data could be obtained free of charge via www.ccdc.cam.ac.uk/conts/retrieving.html (or from the Cambridge Crystallographic Data Centre, 12 Union Road, Cambridge CB2 1EZ, UK; Fax: (+44)-1223-336-033; or deposit@ccdc.cam.uk).

### 4.3. Electrochemical Measurements

Cyclic voltammetry measurements were performed with an E2P potentiostat of BASi Epsilon (USA) composed of a measuring block, a Dell Optiplex 320 computer with an installed EpsilonES-USB-V200 program, and a C3 electrochemical cell. A stationary glassy-carbon electrode (with a diameter of 3.0 mm) was used as a working electrode. Ag/AgCl was used as a reference electrode. A platinum wire of 0.5 mm diameter was used as an auxiliary electrode. Measurements were performed under an inert argon atmosphere.

*3,7-Dibenzyl-1,5-di(1′-(R)-phenylethyl)-1,5-diaza-3,7-diphosphacyclooctane* (**3**). 1-(*R*)-phenylethylamine (0.83 g, 6.85 mmol) in dry ethanol (7 mL) was added to a solution of bis(hydroxymethyl)benzylphosphine, obtained by the stirring of the mixture of benzylphosphine (0.85 g, 6.85 mmol) and paraformaldehyde (0.41 g, 13.70 mmol) at 100–110 °C up to homogenization in dry ethanol (8 mL). The reaction mixture was stirred at 70–80 °C for 6 h and cooled. The precipitate formed was filtered off, washed with ethanol, and dried for 5 h at 2 · 10^−2^ torr to give **3** as a white powder (1.07 g, 58%). M.p. 130–132 °C. ^1^H-NMR (CDCl_3_): δ 1.14 (d, ^3^*J*_HH_ = 6.4 Hz, 6H, C*H*_3_), 2.06 (d, ^2^*J*_HH_ = 13.0 Hz, 2H, P-C*H*_2_-Ph), 2.19 (br. d, ^2^*J*_HH_ = 13.0 Hz, 2H, P-C*H*_2_-Ph), 2.91 (d, ^2^*J*_HH_ = 14.8 Hz, 2H, P-C*H*_2_-N), 3.06 (d, ^2^*J*_HH_ = 14.4 Hz, 2H, P-CH_2_-N), 3.27 (br. d, ^2^*J*_HH_ = 14.4 Hz, 2H, P-C*H*_2_-N), 3.69 (br. d, ^2^*J*_HH_ = 14.8 Hz, 2H, P-CH_2_-N), 4.64 (q, ^3^*J*_HH_ = 6.4 Hz, 2H, C*H*(Me)Ph), 7.07 (t, ^3^*J*_HH_ = 7.8 Hz, 2H, H_ar_), 7.10–7.26 (m, 14H, H_ar_), 7.48 (d, ^3^*J*_HH_ = 7.8 Hz, 4H, H_ar_). ^31^P{^1^H} NMR (CDCl_3_): δ_P_ -67.54 (s). Anal. Calc. for C_34_H_40_P_2_N_2_ (538.64): C 75.81, H 7.49, N 5.20, P 11.50%. Found: C 75.60, H 7.57, N 5.16, P 11.61%.

*3,7-Dimesityl-1,5-di(p-tolyl)-1,5-diaza-3,7-diphosphacyclooctane* (**5**). **5** was prepared like **3** from mesitylphosphine (0.91 g, 5.99 mmol), paraformaldehyde (0.36 g, 12.00 mmol), and *p*-toluidine (0.64 g, 5.99 mmol). The reaction time was 10 h, and the yield of **5** was 1.05 g (62%). M.p. > 200 °C (decomp.). ^1^H-NMR (CDCl_3_): δ 2.18 (s, 6H, NC_6_H_4_C*H*_3_), 2.34 (s, 6H, C*H*_3Mes_-*p*), 2.53 (s, 12H, C*H*_3Mes_-*o*), 4.43 (br.d, ^2^*J*_HH_ = 15.0 Hz, 4H, P-C*H*_2_-N), 4.51 (dd, ^2^*J*_HH_ = 15.0 Hz, ^2^*J*_PH_ = 3.2 Hz,4H, P-C*H*_2_-N), 6.27 (d, ^3^*J*_HH_ = 8.4 Hz, 4H, NC_6_*H*_4_CH_3_-*o*), 6.92 (d, ^3^*J*_HH_ = 8.4 Hz, 4H, NC_6_*H*_4_CH_3_-*m*), 6.96 (s, 4H, H_Mes_). ^31^P{^1^H} NMR (CDCl_3_): δ_P_ −44.41 (s). Anal. Calc. for C_36_H_44_P_2_N_2_ (566.70): C 76.30, H 7.83, N 4.94, P 10.93%. Found: C 76.18, H 7.92, N 4.75, P 10.78%.

*Bis(3,7-Diphenyl-1,5-di(p-methoxyphenyl)-1,5-diaza-3,7-diphosphacyclooctane)-bis(acetonitrile)iron(II)] tetrafluoroborate* (**6**). [Fe(CH_3_CN)_6_](BF_4_)_2_ (0.088 g, 0.19 mmol) in dry acetonitrile (4 mL) was added to **1** (0.200 g, 0.38 mmol) in dry chlorophorm (4 mL). The reaction mixture was stirred at room temperature for 12 h. The ligand was completely dissolved, and the reaction mixture turned dark-red. The reaction mixture was concentrated to ca. 3 mL under reduced pressure, the precipitate was filtered off, washed with diethyl ether, and dried for 3 h at 2·10^−2^ torr to give complex 6 as a brown powder (0.112 g, 43%). Decomposed at 160 °C. ^1^H-NMR (CD_3_CN): δ 1.96 (s, 6H, C*H*_3_CN), 3.62 (s, 6H, -OC*H*_3_), 3.74 (d, ^2^*J*_HH_ = 6.5 Hz, 4H, P-C*H*_2_-N), 3.79 (m, 4H, P-C*H*_2_-N), 3.88 (s, 6H, -OC*H*_3_), 4.06–4.11 (m, 4H, P-C*H*_2_-N), 4.32–4.35 (m, 4H, P-C*H*_2_-N), 6.68 (d, ^3^*J*_HH_ = 9 Hz, 4H, N-C_6_*H*_4_OCH_3_-*o*), 6.78 (d, ^3^*J*_HH_ = 8 Hz, 4H, N-C_6_*H*_4_OCH_3_-*o*), 6.80 (d, ^3^*J*_HH_ = 8 Hz, 4H, N-C_6_*H*_4_OCH_3_-*m*), 6.97 (d, ^3^*J*_HH_ = 9 Hz, 4H, N-C_6_*H*_4_OCH_3_-*m*), 7.51–7.55 (m, 4H, *H*_ar_), 7.70 (br s, 7H, *H*_ar_), 7.82 (br s, 9H, *H*_ar_). ^31^P{^1^H} NMR (CD_3_CN): δ_P_ 30.52–31.46 (m), 34.23–35.20 (m). ESI MS: *m*/*z* 542 [6-2 BF_4_-2 CH_3_CN]^2+^, 1103 [6-2 BF_4_-2 CH_3_CN + H_2_O]^+^. Anal. Calc. for C_64_H_70_P_4_N_6_O_4_FeB_2_F_8_ (1340.63): C 57.34, H 5.26, N 6.27, P 9.24%. Found: C 57.03, H 5.41, N 6.49, P 9.08%.

*[Bis(3,7-Dibenzyl-1,5-diphenyl-1,5-diaza-3,7-diphosphacyclooctane)-bis(acetonitrile)iron(II)] tetrafluoroborate* (**7**). [Fe(CH_3_CN)_6_](BF_4_)_2_ (0.0953 g, 0.215 mmol) in dry acetonitrile (6 mL) was added to a suspension of **2** (0.2081 g 0.43 mmol) in dry acetonitrile (70 mL) at 70 °C. The reaction mixture was stirred at 70 °C for 3.5 h and then at room temperature for 12 h. The ligand was completely dissolved, and the reaction mixture turned yellow-orange. The reaction mixture was concentrated to ca. 25 mL under reduced pressure, and the residue was crystallized at −15 °C for 24 h. The precipitate was filtered off, washed twice with cold acetonitrile and then with diethyl ether, and dried for 3 h at 2·10^−2^ torr to give complex **7** as yellow crystals (0.110 g, 40%). M.p. > 200 °C (decomp.)

^1^H-NMR (CD_3_CN): δ 1.96 (s, 6H, C*H*_3_CN), 2.39 (d, ^2^*J*_HH_ = 12.8 Hz, 2H, P-C*H*_2_-Ph), 2.78 (d, ^2^*J*_HH_ = 13.6 Hz, 2H, P-C*H*_2_-Ph), 3.32–3.68 (m, 10H, P-C*H*_2_-Ph + P-C*H*_2_-N), 3.96–4.24 (m, 8H, P-C*H*_2_-N), 4.34–4.44 (m, 2H, P-C*H*_2_-N), 6.50 (d, ^3^*J*_HH_ = 8.0 Hz, 4H, N-C_6_*H*_5_-*o*), 6.56 (br d, ^3^*J*_HH_ = 7.2 Hz, 4H, N-C_6_*H*_5_-*o*), 6.83 (t, ^2^*J*_HH_ = 7.8 Hz, 2H, N-C_6_*H*_5_-*p*), 7.02 (br d, ^3^*J*_HH_ = 8.4 Hz, 4H, P-CH_2_- C_6_H_5_-*o*), 7.07–7.15 (m, 6H, *H*_ar_), 7.19–7.28 (m, 6H, *H***_a_**_r_), 7.42–7.49 (m, 6H, *H***_a_**_r_), 7.54–7.64 (m, 8H, *H***_a_**_r_). ^31^P{^1^H} NMR (CD_3_CN): δ_P_ 31.44 (m), 35.57 (m). MALDI MS: *m*/*z*: 1019.7 [**7**-2 BF_4_-2 CH_3_CN]^+^, 1050.7 [**7**-2BF_4-_2CH_3_CN-H + 2O]^+^. IR (Nujol, ν, cm^−1^): 1060 vs. br. (BF_4_). Anal. Calc. for C_64_H_70_P_4_N_6_FeB_2_F_8_ (1276.63): C 60.21, H 5.53, N 6.58, P 9.70%. Found: C 60.06, H 5.64, N 6.71, P 9.56%.

*[Bis(3,7-Dibenzyl-1,5-di(1′-(R)-phenylethyl)-1,5-diaza-3,7-diphosphacyclooctane)-bis(acetonitrile)iron(II)] tetrafluoroborate* (**8**). [Fe(CH_3_CN)_6_](BF_4_)_2_ (0.0371 g, 0.085 mmol) in dry acetonitrile (1 mL) was added to a solution of **3** (0.0904 g, 0.17 mmol) in 10 mL dry acetonitrile. The reaction mixture was stirred at room temperature for 12 h and turned yellow-brown. The reaction mixture was concentrated to ca. 3 mL under reduced pressure, and the diethyl ether (5 mL) was added to precipitate the product. The precipitate was filtered off, washed twice with diethyl ether, and dried for 3 h at 2·10^−2^ torr to give complex 8 as a yellow powder (0.131 g, 56%). Decomposed at 140 °C.

^1^H-NMR (CD_3_CN): δ 0.55 (d, ^3^*J*_HH_ = 6.6 Hz, 6H, CH(*Me*)Ph), 1.74 (d, ^3^*J*_HH_ = 7.2 Hz, 6H, CH(*Me*)Ph), 1.96 (s, 6H, C*H*_3_CN), 2.14–2.29 (m, 8H, P-C*H*_2_-Ph), 2.81–2.85 (m, 2H, P-C*H*_2_-N), 2.96–3.02 (m, 2H, P-C*H*_2_-N), 3.09–3.19 (m, 4H, P-C*H*_2_-N), 3.24–3.32 (m, 4H, P-C*H*_2_-N + C*H*(Me)Ph), 3.62 (d.m, ^2^*J*_HH_ = 15.0 Hz, 2H, P-C*H*_2_-N), 3.74 (br.d, ^2^*J*_HH_ = 15.0 Hz, 2H, P-C*H*_2_-N), 3.88 (m, 2H, P-C*H*_2_-N), 4.18 (q, ^3^*J*_HH_ = 7.2 Hz, 2H, C*H*(Me)Ph), 6.69 (m, 4H, *H*_ar_), 7.07–7.45 (m, 36H, *H*_ar_). ^31^P{^1^H} NMR (CD_3_CN): δ_P_ 25.61 (m), 34.19 (m). ESI_pos_ MS: *m*/*z* 1208 [**8**-2 BF_4_-CH_3_CN + F + O]^+^, 1192 [**8**-2 BF_4_-CH_3_CN + F]^+^. IR (Nujol, ν, cm^−1^): 1060 vs. br. (BF_4_). Anal. Calc. for C_72_H_86_N_6_P_4_FeB_2_F_8_ [1388.84]: C 62.27, H 6.24, N 6.05, P 8.92%. Found: C 61.96, H 6.40, N 6.36, P 8.74%.

***[****Bis(3,7-Di(pyridine-2′-yl)-1,5-di(p-tolyl)-1,5-diaza-3,7-diphosphacyclooctane)-bis(acetonitrile)iron(II)] tetrafluoroborate* (**9**). **9** as an orange powder was obtained like **8** from 4 (0.0985 g, 0.203 mmol) and [Fe(CH_3_CN)_6_](BF_4_)_2_ (0.0447 g, 0.101 mmol), except that 4 was used as a suspension. The yield of 9 was 0.079 g (61%). Decomposed at 160 °C. ^1^H-NMR (CD_3_CN): δ 1.96 (s, 6H, C*H*_3_CN), 2.11 (s, 6H, NC_6_H_4_C*H*_3_), 2.32 (s, 6H, NC_6_H_4_C*H*_3_), 2.90–2.99 (m, 2H, P-C*H*_2_-N), 3.28 (br.d, ^2^*J*_HH_ = 12.8 Hz, 2H, P-C*H*_2_-N), 3.34 (br.d, ^2^*J*_HH_ = 14.2 Hz, 2H, P-C*H*_2_-N), 3.41 (br.d, ^2^*J*_HH_ = 11.6 Hz, 2H, P-C*H*_2_-N), 3.73–3.83 (m, 2H, P-C*H*_2_-N), 4.13 (br.d, ^2^*J*_HH_ = 14.2 Hz, 2H, P-C*H*_2_-N), 4.24–4.33 (m, 2H, P-C*H*_2_-N), 4.38–4.48 (m, 2H, P-C*H*_2_-N), 6.52 (d, ^3^*J*_HH_ = 8.6 Hz, 4H, NC_6_*H*_4_CH_3_-*o*), 6.71 (d, ^3^*J*_HH_ = 8.4 Hz, 4H, NC_6_*H*_4_CH_3_-*o*), 6.91 (d, ^3^*J*_HH_ = 8.6 Hz, 4H, NC_6_*H*_4_CH_3_-*m*), 6.95 (m, 2H, *H*^4^_pyr_), 7.11 (d, ^3^*J*_HH_ = 8.4Hz, 4H, NC_6_*H*_4_CH_3_-*m*), 7.36 (m, 2H, H^4^_pyr_), 7.51 (m, 2H, *H*^5^_pyr_), 7.70 (m, 2H, *H*^5^_pyr_), 7.95 (br.d, ^3^*J*_HH_ = 8.0 Hz, 2H, *H*^6^_pyr_), 8.13 (br.d, ^3^*J*_HH_ = 4.4 Hz, 2H, *H*^3^_pyr_), 8.16 (m, 2H, *H*^6^_pyr_), 9.03 (br.d, ^3^*J*_HH_ = 3.6 Hz, 2H, *H*^3^_pyr_). ^31^P{^1^H} NMR (CD_3_CN): δ_P_ 34.69 (m), 41.13 (m). ESI_pos_ MS: *m*/*z* 512.4 [9-2 BF_4_-2 CH_3_CN]^2+^, 1059.6 [9-2 BF_4_- 2 CH_3_CN + F + O]^+^. IR (Nujol, ν, cm^−1^): 1060 vs. br. (BF_4_). Anal. Calc. for C_60_H_66_N_10_P_4_FeB_2_F_8_ [1280.58]: C 56.27, H 5.19, N 10.94, P 9.67%. Found: C 56.09, H 5.31, N 11.17, P 9.49%.

*[(3,7-Dimesityl-1,5-di(p-tolyl)-1,5-diaza-3,7-diphosphacyclooctane)tetrakis-(acetonitrile)iron(II)] tetrafluoroborate* (**10**). [Fe(CH_3_CN)_6_](BF_4_)_2_ (0.240 g, 0.53 mmol) in dry acetonitrile (2 mL) was added to a solution of **5** (0.300 g, 0.53 mmol) in dry acetonitrile (6 mL). The reaction mixture was stirred at room temperature for 48 h. The ligand was completely dissolved, and the reaction mixture turned brown. The reaction mixture was concentrated to ca. 3 mL under reduced pressure, and the residue was crystallized at −15 °C for 12 h. The precipitate was filtered off, washed with diethyl ether, and dried for 3 h at 2 · 10^−2^ torr to give complex **10** as a dark-red powder (0.116 g, 28%). Decomposed at 130 °C. ^1^H-NMR (CD_3_CN): δ 2.18 (s, 6H, NC_6_H_4_C*H*_3_), 2.38 (s, 6H, C*H*_3Mes_-*p*), 2.44 (s, 12H, C*H*_3Mes_-*o*), 4.77–4.82 (m, 4H, P-C*H*_2_-N), 4.93–4.97 (m, 4H, P-C*H*_2_-N), 6.54 (br s, 4H, NC_6_*H*_4_CH_3_-*o*), 6.99 (br.d, ^3^*J*_HH_ = 6 Hz, 4H, NC_6_*H*_4_CH_3_-*m*), 7.17 (s, 4H, *H*_Mes_). ^31^P{^1^H} NMR (CD_3_CN): δ_P_ 55.56. MALDI MS: *m*/*z* 621 [**10**-2 BF_4_-4 CH_3_CN]^+^. ESI_pos_ MS: *m*/*z* 621 [**10**-2 BF_4_-4 CH_3_CN]^+^. IR (Nujol, ν, cm^−1^): 1059 vs. br. (BF_4_). Anal. Calc. for C_44_H_56_N_6_P_2_FeB_2_F_8_ [960.36]: C 55.03, H 5.88, N 8.75, P 6.45%. Found: C 54.87, H 6.01, N 8.97, P 6.28%.

*[Bis(3,7-Dibenzyl-1,5-di(1′-(R)-phenylethyl)-1,5-diaza-3,7-diphosphacyclooctane)- tetrafluoroboratoiron(II)] tetrafluoroborate* (**11b**). [Fe(H_2_O)_6_](BF_4_)_2_ (0.0314 g, 0.095 mmol) was added to a solution of **3** (0.100 g, 0.19 mmol) in dry acetone (10 mL). The reaction mixture was stirred at room temperature for 12 h and turned dark-red. In ^31^P-NMR spectrum, two pairs of doublets were observed at 38.21 and 31.03 ppm (^2^*J*_PP_ ≈ ^2^*J*_PP_ ≈ 70.7 Hz) (**11a**), 37.30 and 29.94 ppm (^2^*J*_PP_ ≈ ^2^*J*_PP_ ≈ 71.8 Hz) (**11b**) with a ratio of intensities of 1.4:1. The reaction mixture was concentrated to ca. 2 mL under reduced pressure, and the residue was crystallized at −15 °C for 12 h. The precipitate was filtered off, washed with cold acetone, and dried for 3 h at 2·10^−2^ torr to give complex 11b as dark-red small crystals (0.054 g, 44%). M.p. > 140 °C (decomp.) ^1^H-NMR (acetone-d_6_): δ 1.10 (d, ^3^*J*_HH_= 6.8 Hz, 6H, CH(*Me*)Ph), 1.58 (d, ^3^*J*_HH_ = 6.8 Hz, 6H, CH(*Me*)Ph), 1.84 (br. d, ^2^*J*_HH_ = 14.2 Hz, 2H, P-C*H*_2_-Ph), 1.94–1.99 (m, 2H, P-C*H*_2_-Ph), 2.44–2.52 (m, 2H, P-C*H*_2_-N), 2.65 (br d, 2H, ^2^*J*_HH_ = 14.2 Hz, P-C*H*_2_-Ph), 2.91–2.98 (m, 2H, P-C*H*_2_-N), 3.04–3.11 (m, 4H, P-C*H*_2_-N), 3.22 (br d, ^2^*J*_HH_ = 13.2 Hz, 2H, P-C*H*_2_-N), 3.54–3.63 (m, 2H, P-C*H*_2_-N), 3.68 (q, ^3^*J*_HH_ = 6.8 Hz, 2H, C*H*(Me)Ph), 3.84–3.94 (m, 4H, P-C*H*_2_-N), 4.01 (m, 2H, P-C*H*_2_-N), 4.14 (q, ^3^*J*_HH_ = 6.8 Hz, 2H, C*H*(Me)Ph), 6.74–6.79 (m, 4H, *H*_a__r_), 6.85–6.90 (m, 4H, *H*_a__r_), 7.10–7.55 (m, 32H, *H*_a__r_). ^31^P{^1^H} NMR (acetone-d_6_): δ_P_ 37.30 (dd, ^2^*J*_PP_ = ^2^*J*_PP_ = 71.8 Hz), 29.94 (dd, ^2^*J*_PP_ = ^2^*J*_PP_ = 71.8 Hz). MALDI MS: *m*/*z* 1132 [11b-2 BF_4_]^+^, 1162 [11b-BF_4_-3 F]^+^. IR (Nujol, ν, cm^−1^): 1057 vs. br. (BF_4_). Anal. Calc for C_68_H_80_N_4_P_4_FeB_2_F_8_ [1306.74]: C 62.50, H 6.17, N 4.29, P 9.48%. Found: C 62.39, H 6.31, N 4.18, P 9.23%. 

*[(Cyclopentadienyl)carbonyl(3,7-dibenzyl-1,5-di(1′-(R)-phenylethyl)-1,5-diaza-3,7-diphosphacyclooctane)iron(II)] tetrafluoroborate* (**12**)**.** A solution of **3** (0.1036 g, 0.192 mmol) in dry toluene (10 mL) was added to solid [CpFe(CO)_2_(THF)]BF_4_ (0.0646 g, 0.192 mmol). The reaction mixture was stirred at room temperature for 24 h and turned yellow-orange. The small amount of a sticky dark resin was removed by the centrifugation, the volatiles were removed from the filtrate under reduced pressure, and the residue was crystallized from diethyl ether. The precipitate was filtered off, washed twice with diethyl ether, and dried for 3 h at 2·10^−2^ torr to give complex 12 as a yellow powder (0.122 g, 82%). M.p. 177–180 °C (decomp.) ^1^H-NMR (acetone-d_6_): δ 1.19 (d, ^3^*J*_HH_= 7.2 Hz, 3H, CH(*Me*)Ph), 1.44 (d, ^3^*J*_HH_= 6.8 Hz, 3H, CH(*Me*)Ph), 2.58 (d, ^2^*J*_HH_= 13.5 Hz, 1H, P-C*H*_2_-Ph), 2.61 (d, ^2^*J*_HH_= 12.8 Hz, 1H, P-C*H*_2_-Ph), 2.73–2.89 (m, 4H, P-CH_2_-N, partially overlapped with the signal of H_2_O in (CD_3_)_2_C(O)), 3.12 (d, ^2^*J*_HH_= 13.5 Hz, 1H, P-C*H*_2_-Ph), 3.18 (d, ^2^*J*_HH_= 12.8 Hz, 1H, P-C*H*_2_-Ph), 3.53–3.64 (m, 2H, P-C*H*_2_-N), 3.66–3.78 (m, 3H, P-C*H*_2_-N + C*H*(Me)Ph), 4.05 (q, ^3^*J*_HH_= 7.2 Hz, 1H, C*H*(Me)Ph), 5.18 (s, 5H, C_5_*H*_5_), 7.01–7.05 (m, 2H, *H*_ar_), 7.15–7.24 (m, 6H, *H*_ar_), 7.25–7.36 (m, 12H, *H*_ar_). ^31^P{^1^H} NMR (acetone-d_6_): δ_P_ 55.55 (d, ^2^*J*_PP_= 116.4 Hz), 56.27 (d, ^2^*J*_PP_= 116.4 Hz). ESI_pos_ MS: *m*/*z* 687.3 [12 -BF_4_]^+^. IR (KBr, ν, cm^−1^): 1047 vs. br. (BF_4_), 1940 s (CO), 1951 s (CO). Anal. Calc for C_40_H_45_N_2_OP_2_FeBF_4_ [744.40]: C 62.04, H 5.86, N 3.62, P 8.00%. Found: C 61.91, H 5.97, N 3.58, P 7.68%.

*[(Cyclopentadienyl)carbonyl(3,7-di(pyridine-2′-yl)-1,5-di(p-tolyl)-1,5-diaza-3,7-diphosphacyclooctane)iron(II)] tetrafluoroborate* (**13**). A solution of 4 (0.0656 g, 0.135 mmol) in dry toluene (5 mL) was added to solid [CpFe(CO)_2_(THF)]BF_4_ (0.0456 g, 0.136 mmol) in dry toluene (1 mL). The reaction mixture was stirred at 60–70 °C temperature for 4 days and turned yellow-orange. The formation of a noticeable amount of a sticky dark resin was observed. The resin was removed by the centrifugation, the volatiles were removed from the filtrate under reduced pressure, and the residue was crystallized from diethyl ether. The precipitate was filtered off, washed twice with diethyl ether, and dried for 3 h at 2 · 10^−2^ torr to give complex 13 as a yellow powder (0.042 g, 9%). Decomposed at 160 °C. ^1^H-NMR (acetone-d_6_): δ 2.24 (s, 3H, *CH*_3_), 2.28 (s, 3H, *CH*_3_), 4.04 (d, ^2^*J*_HH_= 13.6 Hz, 2H, P-C*H*_2_-N), 4.22 (dm, ^2^*J*_HH_ = 13.6 Hz, ^2^*J*_PH_ ≈ ^4^*J*_HH_ ≈ 3.4 Hz, 2H, P-C*H*_2_-N), 4.44 (dm, ^2^*J*_HH_ = 14.0 Hz, ^2^*J*_PH_ ≈ ^4^*J*_HH_ ≈ 3.4 Hz, 2H, P-C*H*_2_-N), 4.58 (d, ^2^*J*_HH_= 14.0 Hz, 2H, P-C*H*_2_-N), 5.05(t, ^3^*J*_PH_= 1.4 Hz, 5H, C_5_*H*_5_), 7.10 (d, ^3^*J*_HH_ = 9.0 Hz, 2H, NC_6_*H*_4_CH_3_-*o*), 7.15 (d, ^3^*J*_HH_ = 9.0 Hz, 2H, NC_6_*H*_4_CH_3_-*m*), 7.20 (d, ^3^*J*_HH_ = 8.8 Hz, 2H, NC_6_*H*_4_C*H*_3_-*o*), 7.29 (d, ^3^*J*_HH_ = 8.8 Hz, 2H, NC_6_*H*_4_CH_3_-*m*), 7.64 (ddm, ^3^*J*_HH_ = 7.2 Hz, ^3^*J*_HH_ = 4.8 Hz, 2H, *H*^4^_pyr_), 8.06–8.15 (m, 4H, *H*^5^_pyr_ + *H*^6^_pyr_), 8.91 (d, ^3^*J*_HH_ = 4.8 Hz, 2H, *H*^3^_pyr_). ^31^P{^1^H} NMR (acetone-d_6_): δ_P_ 59.14. ESI_pos_ MS: *m*/*z* 633.4 [12 -BF_4_]^+^. IR (KBr, ν, cm^−1^): 1050 vs. br. (BF_4_), 1940 s (CO), 1950 s (CO). Anal. Calc for C_34_H_35_N_4_OP_2_FeBF_4_ [720.27]: C 56.70, H 4.90, N 7.78, P 8.60%. Found: C 56.58, H 5.07, N 4.79, P 8.43%.

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
