# Peer review of "Synthesis and Structure of Iron (II) Complexes of Functionalized 1,5-Diaza-3,7-Diphosphacyclooctanes"

_molecules, 2020, doi:10.3390/molecules25173775_

Round 1

Reviewer 1 Report

Molecules-893777

This i s a fine chemistry and is an interesting research carried out properly though its presentation may need some polishing.

Thus I suggest acceptance after series of minor changes as suggested here under.

Authors should take care of consistency checking at following Pages/Lines:

P1/L13 "unbulky phenyl" (and similarly at P8/L261) expression I find somewhat deliberate. I admittedly have some aversion towards negative definitions. Thus attribute “unbulky” is problematic for at least two reasons. One that, as in all non-positive definitions, this does not defines a property, instead it says what is not. Second that it implies relations which are not defined clearly. One could well assume that a Ph-group is bulky against a Me- or -Et-group, say. From the paper it turns out then that Authors use this epithet in relation to mesityl groups.

Instead “unbulky” may I suggest using a simple “flat” attribute for these (phenyl, tolyl etc.) substituents. Flat has the connotation being not only planar but it suggests the “simple” meaning, too. Either way I guess it to be closer to what Authors mean, but they have the rights to decide for the final wording.

Before going into further I find it rather unfortunate that Authors did not describe the crystal structure 11b referring to certain "peer-review standards". Such standards undoubtedly exists for some crystallography-specific periodicals where generally high accuracy is desirable. Nevertheless this work is primarily about chemistry and one must recognize that many of the small - molecule goodness criteria would not fit for systems that are inherently more similar to large-molecule crystals (such as proteins and viruses) in their restricted scattering power and resolution, high solvent content and extensive disorder. Thus a cautious discussion is due here as well, as if one would treat a protein crystal structure with restricted information content.

From P5/L159 on the structural features from a crystal of 11b are to be discussed. Some parts of this discussion would request that the underlying crystallographic information (such as the space group and cell axes, raw experimental data numbers and model fitting results i.e. R values, Flack x) be provided.

Otherwise basic molecular features (such as the twofold molecular symmetry thus shape and like) will not be understood and remain obscure.

Also sentences at P5/L173-174 must be corrected such that”... on the axe 2 (atoms Fe and B lie on axis 2)” should be “on a twofold axis (atoms Fe and B lie on the twofold)”.

At L175 the sentence's ending must be “molecules could not be identified”. (I am sure that there might have been more than two water and two acetone solvents in the huge channels parallel to the trigonal axes. This possibility is clear from the packing scheme of the 11b crystal TUWHIF.)

P8/L281-285: CHECK pls. "Daltonic" or "Daltonics"?

P9/L307: F2 should print F 2 superscript(square)

P9/L309: instead "refined as riding" write "treated as riding" (as riding is NOT refinement)

P9/L310: instead "determined on the basis of" write "confirmed by"

P9/L312-315 Please correct and check that italicizing and style conforms to crystallographic standards (such as space group name P italicized, R1 subscript 1, Rw subscript w)

Author Response

The reviewer’s 1 remarks and authors responses:

(please take into account that the use of "Track Changes" function in Microsoft Word led to the changes in line numbers in comparison with the previous variant)

We are very thankful to the reviewer for a very careful peer review of our manuscript.

The reviewers report

This i s a fine chemistry and is an interesting research carried out properly though its presentation may need some polishing.

Thus I suggest acceptance after series of minor changes as suggested here under.

Authors should take care of consistency checking at following Pages/Lines:

  • P1/L13 "unbulky phenyl" (and similarly at P8/L261) expression I find somewhat deliberate. I admittedly have some aversion towards negative definitions. Thus attribute “unbulky” is problematic for at least two reasons. One that, as in all non-positive definitions, this does not defines a property, instead it says what is not. Second that it implies relations which are not defined clearly. One could well assume that a Ph-group is bulky against a Me- or -Et-group, say. From the paper it turns out then that Authors use this epithet in relation to mesityl groups.

Instead “unbulky” may I suggest using a simple “flat” attribute for these (phenyl, tolyl etc.) substituents. Flat has the connotation being not only planar but it suggests the “simple” meaning, too. Either way I guess it to be closer to what Authors mean, but they have the rights to decide for the final wording.

The response:

The word “unbulky” was replaced by the “relatively small” (abstract P1, L13-14; discussion P9, L-307).

  • Before going into further I find it rather unfortunate that Authors did not describe the crystal structure 11b referring to certain "peer-review standards". Such standards undoubtedly exists for some crystallography-specific periodicals where generally high accuracy is desirable. Nevertheless this work is primarily about chemistry and one must recognize that many of the small - molecule goodness criteria would not fit for systems that are inherently more similar to large-molecule crystals (such as proteins and viruses) in their restricted scattering power and resolution, high solvent content and extensive disorder. Thus a cautious discussion is due here as well, as if one would treat a protein crystal structure with restricted information content.

From P5/L159 on the structural features from a crystal of 11b are to be discussed. Some parts of this discussion would request that the underlying crystallographic information (such as the space group and cell axes, raw experimental data numbers and model fitting results i.e. R values, Flack x) be provided.

Otherwise basic molecular features (such as the twofold molecular symmetry thus shape and like) will not be understood and remain obscure.

The response:

After the additional refinement and treatment of X-ray data the details of structure of 11b were again included in the experimental part of text (P11, lines 365-382) and the redeposited CIF-file was submitted. .

  • Also sentences at P5/L173-174 must be corrected such that”... on the axe 2 (atoms Fe and B lie on axis 2)” should be “on a twofold axis (atoms Fe and B lie on the twofold)”.

The response:

The sentence at P5 (now it is P7, L 203-204) was corrected to “The molecule 11b in this crystal is on the special position on the twofold axe 2 (atoms Fe and B lie on the twofold axis 2)”.

  • At L175 the sentence's ending must be “molecules could not be identified”. (I am sure that there might have been more than two water and two acetone solvents in the huge channels parallel to the trigonal axes. This possibility is clear from the packing scheme of the 11b crystal TUWHIF.)

The response:

The words “molecules were not found” were replaced by “molecules could not be identified” (now P7, Lines 213-214). Also “acetone” was added to the possible unidentified molecules (P7, L218)

  • P8/L281-285: CHECK pls. "Daltonic" or "Daltonics"?

The response:

In the part “Materials and Methods” (now P11, L328-332) “Daltonic” was everywhere replaced by "Daltonics"

  • P9/L307: F2 should print F 2 superscript(square)

The response:

F2 was replaced by F 2 (superscript) (now P11, L 354, X-ray crystallography).

  • P9/L309: instead "refined as riding" write "treated as riding" (as riding is NOT refinement)

The response:

The sentence “refined as riding” was replaced by "treated as riding" (now P11, L357, X-ray crystallography).

  • P9/L310: instead "determined on the basis of" write "confirmed by"

The response:

The sentence “determined on the basis of ” was replaced by "confirmed by” (now P11, L357, X-ray crystallography).

  • P9/L312-315 Please correct and check that italicizing and style conforms to crystallographic standards (such as space group name P italicized, R1 subscript 1, Rw subscript w)

The response:

All these remarks were taken into account and the corresponding corrections were made (now P 11, L-359-378, the X-ray structures description of 8 and 11b)

Reviewer 2 Report

This paper describes the synthesis of several complexes of Fe with 1,5-diaza-3,7-diphosphacyclooctanes. The structures of two of these complexes 8 and 11b are determined by crystallography. 

The discussion of structure 8 is poor. The dimensions in the coordination sphere should be included (in the text or in a Supplementary Publication). It is stated that the geometry is a strongly distorted octahedron but this is an exaggeration, the word strongly should be omitted. There is no discussion of the dimensions in the coordination sphere. Figure 2 is very poor. It is impossible to work out the structure from the diagram. (This comment also applies to the Figure for 11b) Variations in colour would help but the authors need to find a way of making the structures understandable.

In this figure the atoms in the second ligand are named differently with a prime from those in the cif file which are given the letter B. This is unacceptable. The prime is conventionally used for a symmetry related atom as indeed it is for structure 11b and should not be used for unique atoms in the asymmetric unit.

It seems perverse to quote the non-bonded Fe-N5 distances but not the bonded Fe-N distances. It is stated that both ligands have chair-boat conformations but there is no identification given of which rings are so described. Atom labels should be used in the figures and the rings identified in the text.

The treatment of structure 11b in the paper is strange. It is rightly stated that the details of the cation are of an acceptable standard but that there is a problem with a missing anion. The solution adopted by the authors is not to include details of the structure in the text (or Supplementary Publication) but to discuss the structure in detail in the text. This is unacceptable.

 I have used  the submitted cif file to examine the structure in WinGX and it seems obvious that the atom they identify as F30 must be a boron. There are 3 peaks around F30 in the difference Fourier map which can be identified as F atoms. It must be possible to obtain a disordered BF4 anion from this electron density by creating a disordered set of F atoms around this B (formerly F30) though it may be necessary to move the B off the x axis to achieve this. When they have done this successfully the structure can be included in the main text of the paper.  However the discussion of both structures need improvement. the details for the coordination geometry around the metal be included in the text in a Table comparing it with the dimensions of structure 8.

In the further refinement of structure 11b, some features need attention. It is absurd to have a negative value for BASF. C52 (and possibly C51) is disordered as indicated by the thermal parameters which are unreasonable and these atoms should be refined in two positions. The reflection affected by the beamstop should be removed. The chemical formula in the cif file should be corrected. The coordinates of two of the solvent molecules should be given equivalent positions closer to the cation.

In the text the symmetry element described by the prime should be quoted.

Line 180  dihedral angles have standard deviations. Some dimensions need Å and standard deviations

Line 191, the word close is inappropriate and should be removed.

The name is van der Waals not as written

In general, the English in this paper needs considerable improvement.

Author Response

The reviewer’s 2 remarks and authors responses:

(please take into account that the use of "Track Changes" function in Microsoft Word led to the changes in line numbers in comparison with the previous variant)

We are very thankful to the reviewer for a very careful peer review of our manuscript.

The reviewers report

  • This paper describes the synthesis of several complexes of Fe with 1,5-diaza-3,7-diphosphacyclooctanes. The structures of two of these complexes 8 and 11b are determined by crystallography.
    • The discussion of structure 8 is poor. The dimensions in the coordination sphere should be included (in the text or in a Supplementary Publication). It is stated that the geometry is a strongly distorted octahedron but this is an exaggeration, the word strongly should be omitted. There is no discussion of the dimensions in the coordination sphere. Figure 2 is very poor. It is impossible to work out the structure from the diagram. (This comment also applies to the Figure for 11b). Variations in color would help but the authors need to find a way of making the structures understandable.

The response:

The remarks were taken into account. The text of the article was corrected (pages 4-5, lines 108-119, 131-144) the word “strongly” (P4, L110) was omitted and the values of the dimensions of the coordination sphere were included partially in the text (bond angles and lengths, P4, L111-117) and to the Table 1 (Page 8). The pictures 2a and 2b was redone in color, and more clear projections were chosen.

  • In this figure the atoms in the second ligand are named differently with a prime from those in the cif file which are given the letter B. This is unacceptable. The prime is conventionally used for a symmetry related atom as indeed it is for structure 11b and should not be used for unique atoms in the asymmetric unit.

The response:

The remarks were taken into account, the atom labels were corrected both in the text (P 4-5, L 108-119, 131-144) and in the figure 2a. The atom labels of the second ligand were given with the letter B in accordance with the cif-file.

  • It seems perverse to quote the non-bonded Fe-N5 distances but not the bonded Fe-N distances. It is stated that both ligands have chair-boat conformations but there is no identification given of which rings are so described. Atom labels should be used in the figures and the rings identified in the text.

The response:

The bonded Fe-N (40) and Fe-N(43) distances were indicated both in text (P4, L 115-116) and in the Table 1 (P8). The described ligand cycles were identified in the text (P5, L136).

4) The treatment of structure 11b in the paper is strange. It is rightly stated that the details of the cation are of an acceptable standard but that there is a problem with a missing anion. The solution adopted by the authors is not to include details of the structure in the text (or Supplementary Publication) but to discuss the structure in detail in the text. This is unacceptable.

The response

After the additional refinement and treatment of X-ray data the details of structure of 11b were restored in the experimental part of text (P11, lines 366-380) and the redeposited CIF-file was submitted. .

5) I have used  the submitted cif file to examine the structure in WinGX and it seems obvious that the atom they identify as F30 must be a boron. There are 3 peaks around F30 in the difference Fourier map which can be identified as F atoms. It must be possible to obtain a disordered BF4 anion from this electron density by creating a disordered set of F atoms around this B (formerly F30) though it may be necessary to move the B off the x axis to achieve this. When they have done this successfully the structure can be included in the main text of the paper.  

 The response

The asymmetric part of the crystal consists of 1/2 of the cation, solved the BF4 anion in a special position on axis 3 (1/3 of the anion) and to maintain the electroneutrality of the crystal 1/6 of the anion. We tried to solve this anion, and it appears close to the point of intersection of the twofold and trifold axes. As a result, it is disordered and it was not possible to find a tetrahedral anion (see figure). We decided not to specify this anion, but to take into account its presence in the crystal together with non-solved solvent molecules using the SQUIZ procedure in the PLATON program.

Disordering of the BF4 anion in the crystal 11b

6) However the discussion of both structures need improvement. the details for the coordination geometry around the metal be included in the text in a Table comparing it with the dimensions of structure 8.

The response

The Table 1 containing the details of the coordination geometry around the iron for complexes 8 and 11b was included in the text (P8). The comparison of the cation geometries for 8 and 11b was performed and showed more distorted octahedral geometry of 11b (P7, L232-236).

It should be mentioned, that the corrections of the structure description of 11b (P7-8, L219-250) were not correctly shown by “Track Changes" function in Microsoft Word due to some failure, so this text was colored in yellow.

7) In the further refinement of structure 11b, some features need attention. It is absurd to have a negative value for BASF.

The response

The twinning was removed.

8) C52 (and possibly C51) is disordered as indicated by the thermal parameters which are unreasonable and these atoms should be refined in two positions.

The response

Disordering of the C52 atom will lead to a non-planar conformation of the acetone molecule. Therefore, we applied the EADP option to this solvate molecule.

9) The reflection affected by the beamstop should be removed. The chemical formula in the cif file should be corrected. The coordinates of two of the solvent molecules should be given equivalent positions closer to the cation.

The response

Remarks were taken into account, corrections were made in the text of the article (experimental part, P11, lines 366-379) and in the CIF-file.

10) In the text the symmetry element described by the prime should be quoted.

The response

Remarks were taken into account, corrections were made in the text of the article (P7, L221-222)

11) Line 180  dihedral angles have standard deviations. Some dimensions need Å and standard deviations.

The response.

Remarks were taken into account, corrections were made in the text of the article (P4-5, 7-8, the descriptions of X-ray structures)

12) Line 191, the word close is inappropriate and should be removed.

The response.

The words “close” (P5, L142 and P7, L 227-228) were removed and the sentences “in close proximity” were replaced by “in the proximity”.

13) The name is van der Waals not as written.

The response

The name “van der Waals” was corrected (P7, L 229).

14) In general, the English in this paper needs considerable improvement.

The response

The English was improved according to both reviewers remarks.

Round 2

Reviewer 2 Report

This paper has been significantly improved in revision and it is now acceptable.

The english is still very poor. It is unfortunate that the authors have not managed to improve it. 

This manuscript is a resubmission of an earlier submission. The following is a list of the peer review reports and author responses from that submission.

Round 1

Reviewer 1 Report

The manuscript of Spiridonova and co-authors contains two X-ray crystal structures for which I obtained the deposited data from the CCDC.

The data for compound 8 have rather bad quality. They can be included in a
publication if the authors describe their procedures in detail. Which
restraints and constraints were used? Why is the merging R-value (R_int) so
high? Did they observe crystal decomposition due to solvent evaporation? Have
reflections been omitted in the refinement, and why?
Note: on line 266 of the manuscript the measurement temperature is given as
100(2) K. Judging the atomic displacement parameters, this is a room-temperature
structure.

The data quality of compound 11b is so poor that they are not suitable for
publication. There is so much noise in the residual electron density that no
chemical conclusions should be drawn from this experiment.

For the two crystal structures the authors have used the SHELXL refinement
program. This software writes the results in two different file formats, the
res-file and the cif-file. Obviously the numbers in both files need to be
identical. For reasons of data integrity, a copy of the res-file is included in
the cif-file.
With compound 11b we see the situation that some numbers in the
cif-data differ from the res-file (displacement parameters of atoms C17, C26,
and C27). In my long career I have never seen such a situation. My only
explanation is that the authors have intentionally falsified their results
in the cif-data. This manipulation was obviously done in order to avoid alerts
in the checkcif-report. (Consequently, the checkcif-report in the Supporting
Information cannot be reproduced from the refined structural model).
This unethical behaviour must result in a rejection of the manuscript!

Author Response

The response to the Reviewer 1.

 Dear reviewer and editors!

The establishment of the structures of the synthesized iron (II) complexes of 1,5-diaza-3,7-diphosphacyclooctanes by the X-ray diffraction was complicated by the low quality of the most of the obtained crystals. In spite of numerous attempts to grow the acceptable crystals under different conditions (from different solvents and their mixtures, from concentrated and diluted solutions, with various cooling rates, by liquid- and gas-phase diffusion methods) as a rule fine powders or highly-mosaic crystals were obtained. It should be mentioned that the complexes appeared to be sensitive to many solvents (especially proton-donating and aproton polar one which contain traces of proton-donating impurities), and this fact also restricted the possibilities of crystallization. So when the grown crystals of 8 and 11b provided more-or –less acceptable diffraction the complete X-ray experiments were performed though the quality of the crystals was relatively low (see fig.1 which presents one of the frames of the experiment for 11b) and the expected data would not be good.

Nevertheless, the structures of compounds 8 and 11b appeared to be possible to solve even on the basis of these data. The R, Rw and Rint show certainly very high values, the geometrical parameters of the molecules have been determined with high errors, but the structures of metal complexes have been determined definitely: the coordination of iron ions has been established, the conformations of 8-membered ligands and the positions of substituents on phosphorus and nitrogen atoms have been also determined unambiguously.

It should be mentioned that the crystals of the complex 11b were grown for several times and two independent experiments with different crystals were performed. Both experiments led to the almost the same results. At first we tried to solve and refine this structure in the space group P3, but the results were worse than in the case of the solving and the refinement in the space group P32, so the final structure of 11b was solved and refined namely in this space group P32. In addition, the crystal of 11b appeared to be both twin and crystal solvate with large cavities containing unfixed solvate molecules. These molecules could not be identified so we had to use SQUEEZE procedure.

Certainly some geometrical parameters of both structures (8 and 11b) were obtained as distorted fragments, for example non-planar phenyl rings, BF4 anions and solvate molecules of acetonitrile (in the crystal of 8) and acetone (in crystal of 11b) have a unrealistic geometry. In order to solve these problems we had to use constrains and restrains (EADP, FLAT, DFIX) in the course of the refinement of these structures.

            So the solving and the refinement of the structure of 11b required long time and numerous attempts. As a result, we had numerous “preliminary” cif-files with similar names and in one of them we corrected thermal parameter of several atoms in order to obtain a better and more clear figure. Unfortunately, during the publication of this structure namely this file with incorrect changes was deposited to CCDC in error. It is a really gross error and we apologize for this mistake. At the present time the structure of 11b has been refined on the basis of the one of the experimental data massive which has provided a little bit better results. The obtained final cif-file was redeposited to CCDC. The revised cif-file of complex 8 was also redeposited to CCDC. The corresponding corrections have been made in the description of X-ray experiment in the experimental part.

            It should be mentioned that X-ray data were used in this article mainly for the confirmation of the principal structures of the synthesized complexes and were in accordance with the data of other physical methods of the structure investigation.

Reviewer 2 Report

This manuscript describes the synthesis of Fe(II) complexes of various diazadiphosphacyclooctane ligands and their characterization by multinuclear NMR spectroscopy (1H and 31P) as well as by MS and IR studies.  Single crystal X-ray diffraction studies were carried out on two of the complexes.

The research and the interpretation of the data appears to be sound but there are some omissions that must be addressed.

1) Raw spectral data (31P and 1H NMR, MS) should be included in the Supplementary material and it is not.  This must be implemented. 

2) Weakly coordinating BF4- groups are unusual but not unknown.  The authors should cite other examples and compare the F...Fe distances with what is observed in other cases (if these are known for Fe).

3) It is a shame that no VT-NMR studies were carried out to investigate the energetics of the conformational changes of the rings.

4) Are the coordinated acetonitrile ligands labile? I did not see any comments on this...

5) As stated in the check-cifs, no details of any absorption correction (if any) are given.

6) While the % of P is given for all reported compounds, the CHN anaylsis is only given for one complex.

7) A number of times (including in the abstract), the charges of the cations are given, but those of the BF4- anions are not. This should be corrected.

Overall the manuscript should be acceptable after the implementation of these recommendations.

Author Response

The response to the Reviewer 2.

Dear reviewer and editors!

We revised the manuscript according to the requirements of reviewer 2 as much as possible.

  • The Electronic Supplentary Material including raw 31P, 1H NMR and mass spectra of all newly synthesized compounds has been formed. It also contains TG/DSC diagram of 11 b
  • We analyzed the described data concerning ВF4 coordination to Fe (II) ions (only examples of terminal F-BF3 coordination were found) and concerning relatively rare chelate F2-BF2 coordination of BF4 to other transition metals (mainly Ag(I)), We compared the metal-F distances and included this comparison into the description of the structure of 11b (page 6).
  • Unfortunately the desirable VT-NMR studies of complexes 6-13 appeared to be very complicated due to their sensitivity to the nature of solvent. All complexes partially but noticeably decomposed in the most convenient solvent for VT-NMR, namely dichloromethane-d2. Complexes 6-10 with acetonitrile co-ligands are stable practically only in acetonitrile where the large excess of CH3CN suppress the processes of co-ligands exchange which probably is caused by the practically inevitable traces of proton-donating impurities in the polar deuterated solvents (in particular acetone-d6 or THF-d8). These complexes are practically insoluble in non-polar solvents which may be dried in sufficient extent. The acetonitrile-d3 provides too narrow temperature range for the VT-NMR experiments and does not allow to observe noticeable changes. Unfortunately the possible temperature range of acetone-d6 was also too narrow for the investigation of complexes 11b -13 which were stable in this solvent; the solubilty of 12 and 13 in toluene was restricted and in addition this solvent show several strong signals of the residual protons in the most important regions. As a result we could not to obtain the VT-NMR data for these complexes.
  • The sensitivity of complexes 6-10 to the traces of proton-donating impurities in solvents indicates that the coordinated acetonitrile ligands are relatively labile and may be substituted. The corresponding text was included in the manuscript (pages 4 and 5).
  • The check-cif files were revised, redeposited and resubmitted.
  • The data of C,H,N –elemental analysis for complexes 6-13 were obtained and included in the corresponding characterization data. As a rule our analytical laboratory avoids to carry out the C,H,N-analysis of fluorine-containing compounds for better safety of their devices, but in the case of an urgent need it can perform it.
  • The charges of BF4- anions were corrected everywhere in the text.

Reviewer 3 Report

The manuscript is well structured with a detailed description of the experimental which is appreciated. The introduction section provides sufficient information on recent literature. The conformational analysis part, based on NMR, should include the corresponding spectra in the Supporting Info, or even better in the manuscript itself (as a separate figure) for one representative complex. I believe that would add to quality of presentation. The crystal structure of 11b shows the correct conformation of the ligand deduced from NMR data, as well as the coordination of BF4 anion, but requires better refinement (R1 and Rw are indeed problematic for a serious publication). In this respect, the amount of acetone in single crystals of 11b could be determined by e.g. TG analysis, supported by DSC. Also, if the crystals keep their integrity upon solvent removal, the authors could try to solve the structure again or try another solvent insted of acetone to grow diffraction-quality single crystals. I suggest major revision at this point.

Author Response

The response to the Reviewer 3.

Dear reviewer and editors!

We revised the manuscript according to the requirements of reviewer 3 as much as possible.

  • We included the NMR spectra of all complexes in Supporting Info (see the answer to the reviewer 2) and the figure 1 presenting 1H NMR spectrum of the complex 9 (as the most representative and less complicated than the spectra of complexes on the basis of P-benzyl substituted ligands) was included in the main text.
  • We have re-grown single crystals of 11b and our colleagues from Kazan Federal University performed their TG/DSC analysis. It showed an endothermic loss of two water molecules and a half of acetone molecule per one molecule of the complex in the temperature range 60 - 138 ºC, (fig. S26 in the Supporting Info) so the crystals additionally contain two undetermined disordered water molecules per one molecule of the complex. The full loss of acetone molecules did not take place up to the decomposition of the complex molecule indicated by the strong exothermic peak at 178-187 ºC. Unfortunately the crystals lost their integrity already at early steps of the solvents removal and their X-ray study was impossible. The corresponding explanations were included in the text of the manuscript after the description of unit cell of 11b (page 5), and the TG/DSC diagram was presented in the Supporting Info (fig. S26).

Round 2

Reviewer 1 Report

In the first version of the paper I detected data manipulation in the X-ray
crystal structure of compound 11b. I cannot see any scientific reason for such
an action. "To obtain a better and more clear figure" as stated by the authors
is definitely not a scientific valid argument. Whether the deposition of the
faked data then happened intentionally or accidentally is not an interesting
question. Manipulation of results is a "no go" at any stage of the analysis.
(It is noteworthy that exactly those numbers were manipulated which lead to
alerts in the automatic structure validation).

According to the cover letter, the crystal of 11b was twinned. This important
information is missing in the experimental description of the structure
determination. I would also expect that the twin law is described in the
manuscript. Also, it is necessary to explain how the twinning was treated in the
intensity integration and in the application of the Squeeze procedure.

Despite serious efforts it can happen that the crystal quality remains low. If
this happens, the solution is not the publication.

Reviewer 3 Report

The authors have implemented all the comments and suggestions, particularly addressing the serious issue raised by the Reviewer 1. In my opinion, the manuscript can now be accepted for publication in Molecules in present form.